# Detection of Serum IgG Specific for *Brachyspira pilosicoli* and “*Brachyspira canis*” in Dogs

**DOI:** 10.3390/vetsci11070302

**Published:** 2024-07-03

**Authors:** Julia Gothe, Matthias Horn, Christoph G. Baums, Romy M. Heilmann, Wieland Schrödl

**Affiliations:** 1Institute of Bacteriology and Mycology, Centre for Infectious Diseases, Faculty of Veterinary Medicine, Leipzig University, 04103 Leipzig, Germany; julia.holthoff@vetmed.uni-leipzig.de (J.G.); schroedl@vetmed.uni-leipzig.de (W.S.); 2Institute for Medical Informatics, Statistics and Epidemiology (IMISE), Faculty of Medicine, University of Leipzig, 04107 Leipzig, Germany; matthias.horn@imise.uni-leipzig.de; 3Department for Small Animals, Veterinary Teaching Hospital, Faculty of Veterinary Medicine, University of Leipzig, 04103 Leipzig, Germany; romy.heilmann@kleintierklinik.uni-leipzig.de

**Keywords:** *Brachyspira pilosicoli*, *Brachyspira canis*, canine intestinal spirochetosis

## Abstract

**Simple Summary:**

This study established laboratory tests for the detection of specific antibodies against bacteria of the genus *Brachyspira* in the sera of dogs. By causing inflammation of the large intestine, distinct species of *Brachyspira* can cause disease in several different animal species, such as pigs and poultry, as well as humans. In dogs, *Brachyspira pilosicoli* and “*Brachyspira canis*” have been isolated from fecal samples and rectal swabs. The former can cause clinical signs, such as diarrhea and reduced growth, in pigs and poultry, as well as reduced egg production in poultry. The latter species is generally considered apathogenic. This study evaluates different antigen preparations of these bacteria for their suitability to detect specific antibodies. We were able to select a suitable antigen preparation and detect specific antibodies against both *Brachyspira* species in individual dogs. These laboratory tests are expected to help indirectly detect infection in dogs in the future and to better understand the role of *Brachyspira* as a putative pathogen in dogs.

**Abstract:**

*Brachyspira pilosicoli* (*B. pilosicoli*) is a pathogen in pigs, poultry, and humans causing colitis, diarrhea, and poor growth rates. Its role as a canine pathogen is controversial, and the seroprevalence of specific IgG antibodies against *B. pilosicoli* in dogs is unknown. A further, not yet officially recognized *Brachyspira* species in dogs is “*Brachyspira canis*” (“*B. canis*“), which is proposed to be apathogenic. This study evaluates enzyme-linked immunosorbent assays (ELISAs) measuring serum IgG antibodies specific for *B. pilosicoli* or “*B. canis*” and investigates levels of specific IgG antibodies against *B. pilosicoli* and “*B. canis*” in a cohort of clinical patients presented at an animal referral clinic. These ELISAs use detergent-extracted antigens from *B. pilosicoli* and “*B. canis*”. To increase analytic specificity, we precipitated the antigens with trichloroacetic acid (TCA) to isolate and concentrate the respective protein fraction. Our results indicate that a large number of serum IgG antibodies bind to shared epitopes of detergent-extracted antigens of the two spirochaetes. Our data also suggest that dogs might not only carry *B. pilosicoli* but also have “*B. canis*”-specific serum IgG antibodies.

## 1. Introduction

*Brachyspira pilosicoli* (*B. pilosicoli*) causes intestinal spirochetosis in pigs, poultry, and humans [1]. Clinical signs in pigs and poultry include diarrhea and loss of production [2]. In pigs, the disease is generally milder than swine dysentery caused by *B. hyodysenteriae*, *B. hampsonii*, or *B. suanatina* [3]. However, experimental infections in pigs and poultry have confirmed the role of *B. pilosicoli* as a pathogen infecting the large intestine [2,4]. Although *B. pilosicoli* has been detected in fecal samples of dogs by different authors, its role as a canine pathogen is less clear as its detection is not consistently associated with diarrhea [5,6,7].

*B. pilosicoli* is a very diverse pathogen undergoing frequent recombination [8]. The outer membrane of *Brachyspira* contains lipooligosaccharides (LOSs) rather than lipopolysaccharides (LPSs) [9]. LOSs have been shown to contribute to the serological heterogenicity of *B. pilosicoli* [10]. It is also known that *B. pilosicoli* expresses diverse outer membrane proteins and lipoproteins [11,12]. Infection with *B. pilosicoli* in pigs causes a systemic IgG response, including antibodies directed against cytoplasmatic antigens and outer membrane proteins [13].

Other *Brachyspira* species are also frequently detected in fecal samples of dogs. However, the taxonomic assignment to a specific species is often not clear. A distinct multilocus enzyme electrophoresis (MLEE) profile assigned canine isolates similar to *B. innocens* as “*B. canis*”, though this is not an accepted species yet [14]. Based on deposited 16S rRNA and *nox* sequences, “*B. canis*” might also be identified via sequencing of these genes [5]. Several studies using cultural investigation suggest that “*B. canis*” is the most common spirochaete of the genus *Brachyspira* in dogs [5,6,15]. It is generally regarded as a commensal rather than a pathogen [16].

Studies on the prevalence of antibodies against *Brachyspira* in dogs have not yet been conducted. Thus, this study evaluated ELISAs designed to detect specific serum IgG antibodies against *B. pilosicoli* and “*B. canis*”.

## 2. Materials and Methods

### 2.1. Brachyspira Strains

*B. pilosicoli* strains 102/06 and 26 and “*B. canis*” strains 284, 313, and S2017 were used for antigen preparation. Strain 26 was originally isolated from a rectal swab of a dog with diarrhea and strains 284 and 313 from rectal swabs of clinically unremarkable dogs from the area of Leipzig, Germany [17]. Strain 102/06 is of porcine origin [18]. Strain S2017 was isolated from a beagle dog living in a kennel in a research facility [17].

### 2.2. Canine Serum Samples

We utilized serum samples from the Small Animal Clinic at the Faculty of Veterinary Medicine of Leipzig University (FVM-UL). These samples were surplus materials initially collected for diagnostic purposes. The local ethics committee at the FVM-UL independently reviewed and granted approval for this study (approval# EK9-2021, approved 31 May 2021). Retrospectively, data on the presence of diarrhea, pre-existing conditions, and presenting complaints were extracted from the patients’ electronic medical records. The gathered data are shown in Appendix A.

### 2.3. Antigen Preparation

Antigen preparation was performed as previously described [17]. Briefly, *Brachyspira* were anaerobically cultivated on blood agar, and the bacterial lawn of each agar plate was rinsed with 2 mL of phosphate-buffered saline (PBS; pH 7.35). After centrifugation, the resulting bacterial pellets were solubilized with a nonionic detergent-based extraction reagent (B-PER bacterial protein extraction reagent; Thermo Fisher Scientific, Dreieich, Germany) following the manufacturer’s instructions. The protein-containing fractions were isolated on the ÄKTAprime plus (Cytiva, Freiburg, Germany) via size-exclusion chromatography using a 5 mL-HiTrap-Desalting Column (Merck, Darmstadt, Germany). After dialysis, the protein concentration was determined using the Advanced Protein Assay Reagent (5× Concentrate, Cycloskeleton Inc., Denver, CO, USA) according to the manufacturer’s instructions.

*B. pilosicoli* antigen was acquired from reference strain 397/1 (102/06) and isolate 26. “*B. canis*” antigen was obtained from isolates 284, 313, and S2017 [17]. Using the same extraction procedure, *Escherichia coli* (*E. coli*) antigen was acquired from isolates of clinically healthy dogs.

### 2.4. Gel Preparation

The detergent-extracted antigens of “*B. canis*”, *B. pilosicoli*, and *E. coli*, obtained as described above, were individually covalently bound to separate CNBr-activated Sepharose 4B (VWR, Dresden, Germany) following the manufacturer’s instructions, yielding a “*B. canis*”, a *B. pilosicoli*, and an *E. coli* gel, respectively.

### 2.5. Enzyme-Linked Immunosorbent Assay with Detergent-Extracted Antigen 

The initial design of the enzyme-linked immunosorbent assay (ELISA) to detect specific IgG antibodies directed against “*B. canis*” and *B. pilosicoli* antigen is shown in Figure 1. ELISA plates were coated with 4 µg/mL detergent-extracted antigen at 4 °C overnight. Columns 1–6 were coated with detergent-extracted “*B. canis*” antigen, while columns 7–12 were coated with detergent-extracted *B. pilosicoli* antigen. Blocking was carried out with 0.1% fish gelatin in PBS (1 h, room temperature). To assess the amount of cross-reacting and specific IgG antibodies, serum preadsorption was conducted on a thermal shaker (1 h; 1400 rpm; 21 °C) separately with the “*B. canis*” gel and the *B. pilosicoli* gel. After preadsorption, the sera were separated from the gel via centrifugation (18,213 g; 5 min). Subsequently, the preadsorbed sera and the sera without preadsorption were each applied in duplicate to immobilized “*B. canis*” and *B. pilosicoli* antigen obtained through detergent extraction (1 h; room temperature). Preadsorbed sera were tested at a final dilution of 1:200, while sera without pre-adsorption were diluted at 1:400. Rabbit-anti-dog IgG (Fc fragment specific) peroxidase-conjugated antibody (Biozol, Eching, Germany) was used as a secondary antibody at a dilution of 1:10,000 and was incubated for one hour at room temperature. Between each incubation period, the ELISA plates were washed thrice with PBS supplemented with 0.1% (*v*/*v*) Tween 20. Plates were developed using 2.2-azino-di-(3-ethylbenzithiazoline sulfonate) (ABTS; Roche, Merck, Darmstadt, Germany) with H_2_O_2_ as the substrate. After 15 min, the optical density (OD) was measured at 405 nm. As neither positive nor negative reference sera were available, the interassay variation was calculated with a randomly selected canine serum included in all nine assays. This interassay standard deviated less than 17% from the respective mean value in any included assay.

### 2.6. Enzyme-Linked Immunosorbent Assay with Detergent-Extracted, Trichloroacetic Acid-Precipitated Antigen

The detergent-extracted antigens of *B. pilosicoli* and “*B. canis*” were precipitated with 10% trichloroacetic acid (TCA) to isolate the protein fraction from other fractions. The precipitation products were washed twice with 80% acetone and 0.9% sodium chloride. Subsequently, the washed protein precipitates were solubilized with 1% SDS, 100 mM Tris/HCl (pH 8,0), and 0,5% NaCl. The isolated products (TCA antigens) were then controlled for specificity in an indirect ELISA using specific antibodies against *B. pilosicoli* and “*B. canis*” that was previously produced through rabbit immunization followed by antibody purification [17]. The TCA antigens were subsequently used in an ELISA to detect specific antibodies against *B. pilosicoli* and “*B. canis*” in canine sera (n = 158). ELISA plates were coated with 1 µg/mL TCA antigen (1 h; room temperature). Columns 1–6 were coated with “B. canis”-TCA antigen, while columns 7–12 were coated with *B. pilosicoli*-TCA antigen. Blocking was performed with 1% casein in PBS (1 h; room temperature). Sera were tested at a final dilution of 1:200 (1 h; room temperature). The optical density (OD) was determined. Rabbit-anti-dog IgG (Fc fragment specific) peroxidase-conjugated antibody (Biozol, Eching, Germany) was used as a secondary antibody at a dilution of 1:5000 and was incubated for one hour at room temperature. Plates were developed using ABTS with H_2_O_2_ as the substrate. After each incubation period, the ELISA plates were washed thrice with PBS supplemented with 0.1% (*v*/*v*) Tween 20. As neither positive nor negative reference sera were available, the interassay variation was calculated using a randomly selected canine serum that was included in all five assays. This interassay standard did not deviate more than 19% from the respective mean value in any included assay.

### 2.7. SDS-Polyacrylamide Gel Electrophoresis (SDS-PAGE)

Two-fold concentrated reducing sample buffer was added 1:2 to all four antigen preparations. Subsequently, these preparations were heated to 95 °C for 5 min. To obtain samples with equal amounts of antigen, antigens were diluted with aqua bidest. to a final concentration of 0.125 mg/mL. A total of 9 µL per sample was then fractionated in a 15% polyacrylamide gel. Subsequently, the gel was stained with Coomassie Blue at room temperature overnight. 

### 2.8. Silver Staining for Lipooligosaccharides

Silver staining originally modified for detection of LPSs was performed as previously described to detect LOSs of *Brachyspira* [19,20]. Slight modifications led to the following staining steps, which were conducted at room temperature on a 3D shaker (low setting). After SDS-PAGE, the gel was washed in 10% ethanol for 10 min. Oxidation of LOSs and other glycoproteins was achieved by incubation in 1% periodic acid for 30 min. Subsequently, the gel was washed twice in 10% ethanol for 10 min. The gel was then fixated overnight with 40% methanol and 10% acetic acid. The next day, the gel was washed twice in 10% ethanol for 10 min and thrice in aqua bidest. for 3 min. Incubation in 0.1% silver nitrate for 30 min was followed by washing in aqua bidest. for 3 min. Finally, the gel was developed via reduction in 4% sodium carbonate solution with 0.2% formaldehyde. After achieving the desired contrast, the color reaction was stopped by exposing the gel to 1% acetic acid.

### 2.9. Statistical Analyses

Differences in median OD values between preadsorbed sera and samples without preadsorption were evaluated using a Friedman omnibus test. For multiple post hoc pairwise comparisons, Dunn–Bonferroni tests were performed. *P*-values smaller than 0.05 were considered statistically significant.

The determination of cut-off values for specific antibodies was based on a standard approach for outlier detection, i.e., we calculated Q3 + 1.5 × IQR, with Q3 and IQR denoting the third quartile and interquartile range, respectively.

The data were analyzed using Prism 9.2.0 statistical software (GraphPad, La Jolla, CA, USA). 

## 3. Results

The primary objective of this study was to evaluate ELISAs for their accuracy in detecting specific IgG antibodies against “*B. canis*” and *B. pilosicoli* antigens.

In the first ELISA, we included preadsorption of the sera with the detergent-extracted antigen of *B. pilosicoli* or “*B. canis*” to assess the level of cross-binding IgG antibodies. Preadsorbed sera and sera without preadsorption (n = 79) were then tested in the ELISA against *B. pilosicoli* and “*B. canis*” detergent-extracted antigen. The results are presented in Figure 2. 

As expected, the original sera (without preadsorption) yielded significantly higher OD values than the respective sera preadsorbed with either *Brachyspira* spp. Sera that were preadsorbed with *B. pilosicoli* antigen yielded significantly lower OD values when tested against *B. pilosicoli* antigen (median OD_405 nm_ = 0.215 [IQR = 0.154]) than the same sera after preadsorption with “*B. canis*” antigen (0.250 [0.160]). Analogously, sera preadsorbed with “*B. canis*” antigen showed significantly lower OD values when tested against “*B. canis*” antigen (0.195 [0.150]) than sera preadsorbed with *B. pilosicoli* antigen (0.223 [0.163]). To test whether antibodies binding to these *Brachyspira* antigens in the ELISA cross-react only between different *Brachyspira* spp., we investigated the effect of preadsorption with *E. coli* detergent-extracted antigen gel for ten randomly selected sera. The results are shown in Figure 3. When tested against *B. pilosicoli* detergent-extracted antigen, the preadsorption of sera with *E. coli* detergent-extracted antigen reduced the median OD_405 nm_ from 1.41 (IQR = 1.42) to 0.316 (0.175). When tested against “*B. canis*” detergent-extracted antigen, the preadsorption of sera with *E. coli* detergent-extracted antigen reduced the median OD_405 nm_ from 1.43 (1.83) to 0.331 (0.262). There was no significant difference in the preadsorption with either *Brachyspira* spp. detergent-extracted antigen gel for these ten sera. These results suggest that the coating antigens used in this ELISA bind IgG antibodies that highly cross-react with very different Gram-negative bacteria.

We, therefore, designed a second ELISA including a different coating antigen to detect more specific IgG antibodies. In the second ELISA, we precipitated the detergent-extracted antigens with TCA before they were used as coating antigens. The specificity of these antigens was confirmed with an indirect ELISA using specific antibodies anti-“*B. canis*” and anti-*B. pilosicoli* generated through rabbit immunization. Only a minor cross-reaction (less than 10%) was detected between the specific anti-*B. pilosicoli* and anti-“*B. canis*” hyperimmune sera and the *B. pilosicoli* and “*B. canis*” TCA antigens (Figure 4).

Since these results indicate specificity, we screened serum samples (n = 158) of a diverse cohort of canine patients in this second ELISA. The results are summarized in Figure 5. As sera from dogs that were specific pathogen-free for *Brachyspira* spp. were not available, outliers were determined via a quantile-based standard approach, yielding the following cut-off values for the ELISA: 0.36 for “*B. canis*” and 0.32 for *B. pilosicoli*. Applying these cut-off values, 17 (10.8%) and 15 (9.5%) of the 158 sera evaluated were seropositive for *B. pilosicoli* and “*B. canis*”, respectively. Ten dogs had distinct levels of antibodies against both *Brachyspira* spp. Sera with distinct levels of antibodies against *B. pilosicoli* are depicted in Figure 5 with a level-based color-coding (also used to indicate the corresponding levels of antibodies against “*B. canis*” antigen). The sera with the highest levels of antibodies against *B. pilosicoli* (orange dots; OD > 0.6) did not have specific antibodies against “*B. canis*”. Dogs with distinct levels of antibodies against *B. pilosicoli* were not documented to have gastrointestinal clinical signs as a presenting complaint. Further information on the presumptive or definitive diagnoses and presenting complaints of these dogs are documented in Appendix A.

Our results indicate that individual dogs might carry specific antibodies against *B. pilosicoli* and/or “*B. canis*”. The antigens used for coating the ELISA plates in this study, however, were not specifically defined and included presumably numerous different detergent-extracted antigens of the two *Brachyspira* spp. We performed SDS-PAGE, followed by silver staining and Coomassie Blue staining, to compare the band profile of the original detergent-extracted antigens with the TCA-precipitated antigens (Figure 6). Silver staining revealed multiple and varying bands in all antigen preparations. Prominent bands at 12–16 kDa for “*B. canis*” detergent-extracted antigens could not be found for *B. pilosicoli*. In both TCA preparations, multiple bands were reduced, especially bands of lower molecular weight ranging from less than 6.5 up to 20 kDa. Coomassie Blue staining also revealed protein bands, mostly of higher molecular weight. Of note, the Coomassie Blue banding pattern was distinct for “*B. canis*” vs. *B. pilosicoli* antigens, but the TCA precipitation did not substantially alter the banding patterns for both *Brachyspira* spp.

## 4. Discussion

Studies on the prevalence of *Brachyspira* spp. in fecal samples of dogs in different countries are available [5,7,17,21,22,23,24]. However, we are not aware of any study investigating the prevalence of antibodies against *Brachyspira* in dogs. This is unlike in pigs, where several research groups have shown that screening of sera for antibodies against *B. hyodysenteriae* is a valuable diagnostic approach to determining the herd status [25,26,27]. Furthermore, specific serum IgG against *B. pilosicoli* was confirmed in vaccinated and challenged pigs using whole-cell *B. pilosicoli* ELISAs [28]. 

We conducted a preadsorption of sera with *B. pilosicoli*, “*B. canis*” and *E. coli* detergent-extracted antigens to validate the specificity of the results obtained in our first ELISA using detergent-extracted antigens of “*B. canis*” and *B. pilosicoli*. As sera of specific pathogen-free or convalescent dogs were not available and experimental infection with *Brachyspira* has not been performed in dogs, a limitation of our study was the lack of positive and negative reference sera. However, the reproducibility of our results was assured through a randomly selected reference serum. The results indicate that a substantial number of serum IgG antibodies detected in this ELISA bind to shared epitopes of bacterial antigens of the two spirochaetes. In pigs, cross-reactive antibodies against *B. hyodysenteriae* and *B. pilosicoli* have been recorded [13]. This cross-reactivity has been shown to produce false-positive ELISA results when whole-cell antigens are used as coating antigens [29]. As the immune response against *B. hyodysenteriae* appears to mainly target LOSs, as shown in immunized and convalescent pigs [30], ELISAs using LOSs as coating antigens can present a valuable alternative [25]. However, some LOSs are serogroup-specific, which may lead to false-negative results [31]. *B. pilosicoli* seems to possess more smooth LOSs which do not necessarily cross-react antigenically with other *B. pilosicoli* strains, eliminating LOSs as a potential antigen for ELISAs [11]. As an alternative to whole-cell or LOSs-ELISAs, recombinant proteins may offer a more specific coating antigen. Still, the genetic and antigenic diversity of *B. pilosicoli* makes it difficult to find a suitable candidate. BmpC, with a molecular weight of 23 kDa, is an immunogenic protein found in *B. pilosicoli* [32]. It was isolated from *B. pilosicoli* strain 95–1000 but could not be identified in other *B. pilosicoli* strains, emphasizing the difficulty of finding a suitable immunogenic protein as an antigen for use in ELISAs designed to detect all *B. pilosicoli* infections [32]. Another putative candidate is the lipoprotein MglB. After infection with *B. pilosicoli*, convalescent pigs produced specific antibodies against this lipoprotein [33]. But again, Southern blot analyses of chromosomal DNA revealed homologous sequences encoding the lipoprotein in strains of *B. innocens*, *B. murdochii*, and *B. alvinipulli* [33]. Based on these published results, we opted to test an extract of detergent-extracted proteins as antigen in an ELISA rather than a single recombinant protein to maximize sensitivity. 

For our second ELISA, we precipitated the detergent-extracted antigens with TCA, which increased the specificity substantially as only minor cross-reactions between anti-*B. pilosicoli* and anti-“*B. canis*” rabbit hyperimmune sera were observed (Figure 4). This may be due to the reduction in unspecific antigens, such as phospholipids and LOSs or the linearization of the proteins through denaturation [34]. Staining of SDS-PAGE with Coomassie Blue revealed no major alterations in the banding pattern before and after precipitation of the antigens with TCA (Figure 6). This confirms that the precipitation did not reduce protein fractions. The silver staining of the detergent-extracted antigens of *B. pilosicoli* and “*B. canis*” before and after precipitation with TCA revealed a reduction in lower molecular weight bands (Figure 6). A previous study on the SDS-PAGE profiles of the aqueous-phase extracts after hot-water phenol extraction of different *B. pilosicoli* strains showed one or more predominant bands at about 16 kDa [10]. This was thought to be the lipid A-core section of LOSs. Varying O-antigen chains cause a ladder-like pattern, common for smooth LOSs [35]. We found three dominant bands between 12–14 kDa in the detergent-extracted antigen of “*B. canis*”. Precipitation of the detergent-extracted “*B. canis*” antigen with TCA reduced these bands to one less-defined band. This suggests that the prominent “*B. canis*” bands between 12–14 kDa are nonproteinaceous antigens such as LOSs. Detergent-extracted *B. pilosicoli* antigen did not show a dominant band in the same region. One might speculate that this indicates that the detergent-based extraction removed LOSs less efficiently from the surface of *B. pilosicoli*. Additionally, extraction methods that do not eliminate proteins and nucleic acids are simply not optimal for the visualization of LPSs [36]. 

Although previous studies on the occurrence of *B. pilosicoli* in rectal swabs and dog feces showed an isolation rate below 5% [5,17,22,24,36], our results now confirm specific antibodies against *B. pilosicoli* to be present in roughly one-tenth of the tested sera (Figure 5). After systemic vaccination and experimental infection of pigs with *B. pilosicoli*, specific IgG titers in serum were highest at the time of bacterial challenge. They declined thereafter, while colonization in the intestinal lumen remained [28]. Nonvaccinated pigs did not develop significant titers to *B. pilosicoli* after experimental infection, despite showing clinical signs of disease and being colonized [28]. We assume that colonization of the canine intestine with *B. pilosicoli* or “*B. canis*” is also not sufficient to induce increased levels of antibodies as determined in the ELISA with the TCA-precipitated, detergent-extracted antigens (e.g., orange dots for *B. pilosicoli;*
Figure 5). It appears reasonable to conclude that these dogs developed increased antibody levels in response to an infection with *Brachyspira*. As “*B. canis*” is considered an apathogenic commensal in dogs, it would seem odd that some dogs had very high levels of IgG antibodies binding to the TCA-precipitated antigen of “*B. canis*”. However, “*B. canis*” is generally shed more frequently by dogs than *B. pilosicoli*, as the isolation rate of “*B. canis*” from rectal swabs ranged from 6% to 8% [5,37]. Intestinal epithelial barrier dysfunction might be associated with increased levels of serum IgG antibodies against commensals [38,39,40]. However, the history of the dogs with increased serum IgG levels against “*B. canis*” did not suggest an underlying chronic inflammatory enteropathy (Appendix A). As only three sera (marked in orange; Figure 5) demonstrated much higher levels of IgG against TCA-precipitated *B. pilosicoli* antigen than against “*B. canis*”, our data overall do not confirm *B. pilosicoli* to be a common canine infectious agent eliciting increased serum IgG levels. Taken together, the results of the TCA-precipitated, detergent-extracted ELISA show serum IgG levels against *B. pilosicoli* and/or “*B. canis*” to be increased in individual dogs. Further studies are thus warranted to investigate the immunogenicity of infection with *Brachyspira* in dogs.

## 5. Conclusions

Extracts of “*B. canis*” and *B. pilosicoli* might include a high degree of antigens recognized by cross-reacting IgG antibodies in canine sera. TCA precipitation of detergent-extracted antigens of “*B. canis*” and *B. pilosicoli* results in a reduction in antigens recognized by cross-reacting antibodies in the described ELISAs. This enables the detection of specific antibodies against “*B. canis*” and *B. pilosicoli* in individual dog sera with the developed ELISA.

## Figures and Tables

**Figure 1 vetsci-11-00302-f001:**
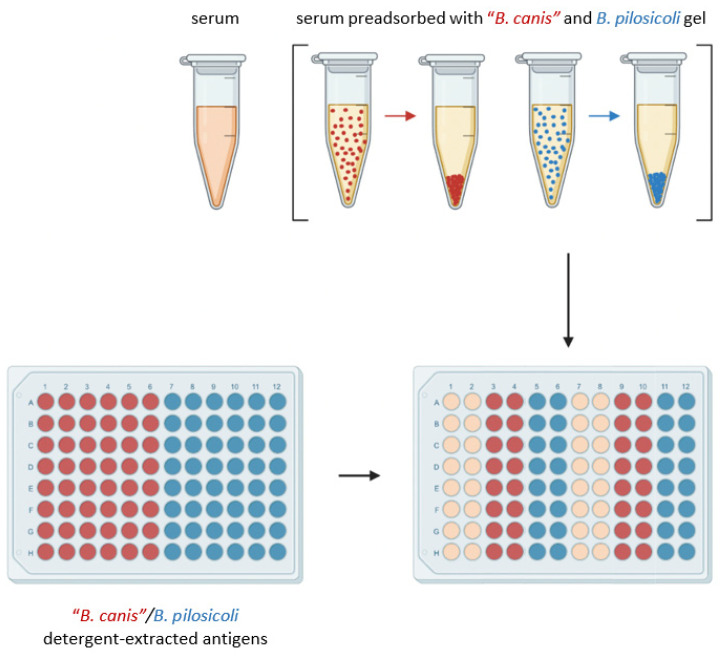
Design of the ELISA measuring IgG antibodies in canine sera that bind to detergent-extracted “*B. canis*” (red dots) and *B. pilosicoli* (blue dots) antigens. Preadsorption of canine sera was conducted with “*B. canis*” (red dots) or *B. pilosicoli* (blue dots) antigen immobilized to a gel as indicated. After centrifugation, supernatants were examined for IgG antibodies against *B. pilosicoli* and “*B. canis*”, respectively. This figure was created with BioRender.com.

**Figure 2 vetsci-11-00302-f002:**
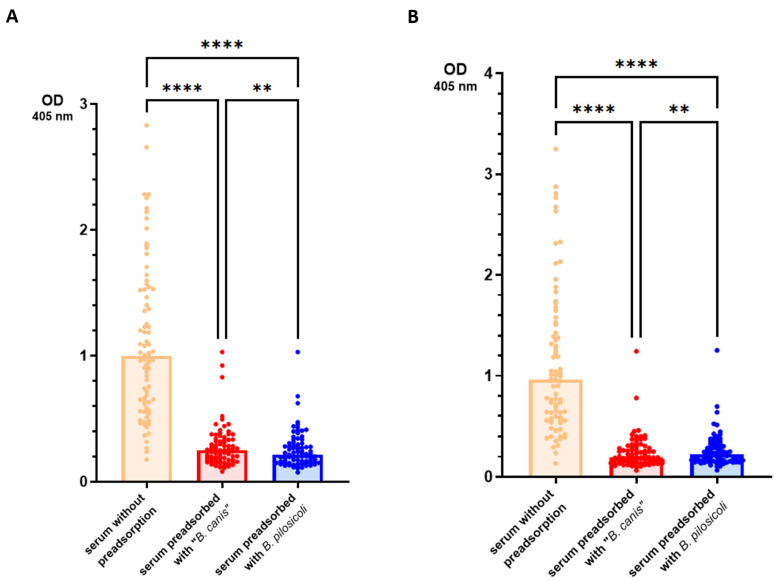
Results of the ELISA measuring IgG antibodies in canine sera that bind to detergent-extracted *B. pilosicoli* (**A**) and “*B. canis*” (**B**) antigens. Sera of 79 dogs with various clinical conditions were tested with and without preadsorption with *B. pilosicoli* and “*B. canis*” antigens. Bars represent median values. Statistical analysis was conducted using a Friedman omnibus test and Dunn–Bonferroni post hoc tests. Statistically significant differences are labeled (**** *p* < 0.0001, ** *p* < 0.01).

**Figure 3 vetsci-11-00302-f003:**
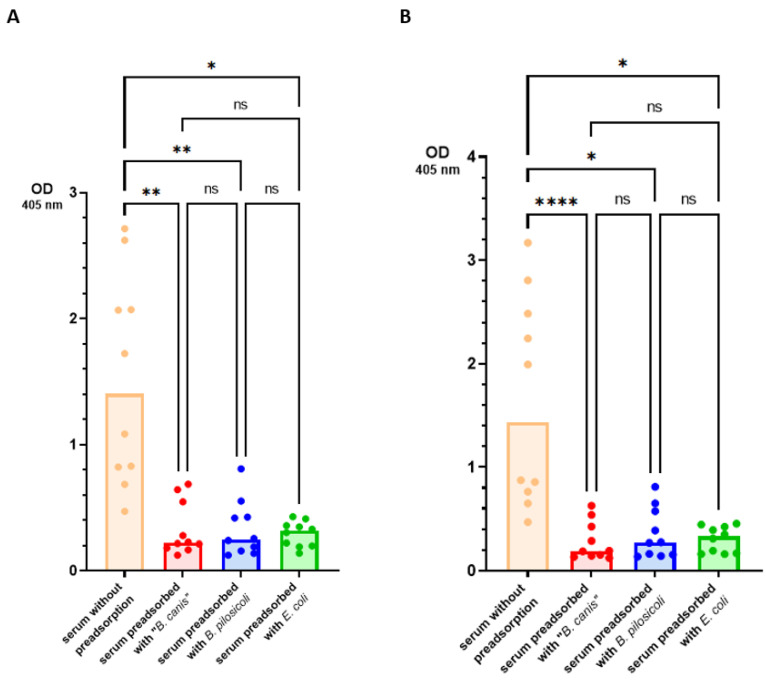
Results of the ELISA conducted to detect specific IgG antibodies against “*B. canis*” and *B. pilosicoli* in canine serum. Preadsorption of 10 canine sera was accomplished with detergent-extracted “*B. canis*”, *B. pilosicoli* and *Escherichia coli* antigen. Sera with and without preadsorption were examined for specific IgG antibodies against detergent-extracted antigens of *B. pilosicoli* (**A**) and “*B. canis*” (**B**). Bars represent median values. Statistical analysis was performed using a Friedman omnibus test and Dunn–Bonferroni post hoc tests. Statistically significant differences are labeled (**** *p* < 0.0001, ** *p* < 0.01, * *p* < 0.05); ns—not significant.

**Figure 4 vetsci-11-00302-f004:**
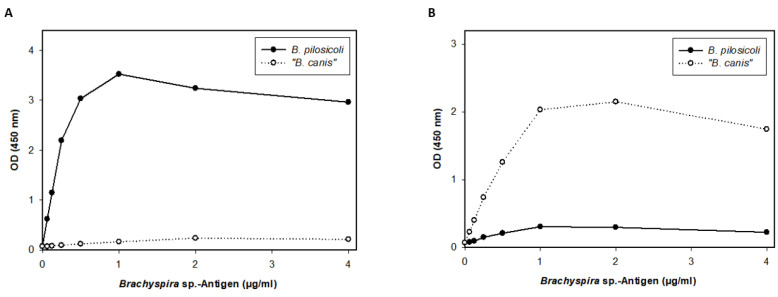
Results of the indirect ELISA including “*B. canis*” and *B. pilosicoli* detergent-extracted antigens precipitated via trichloroacetic acid (TCA) as coating antigen. The indirect ELISA was conducted using specific rabbit anti-*B. pilosicoli* and anti-“*B. canis*” antibodies against *B. pilosicoli* TCA antigen (**A**) and “*B. canis*” TCA antigen (**B**).

**Figure 5 vetsci-11-00302-f005:**
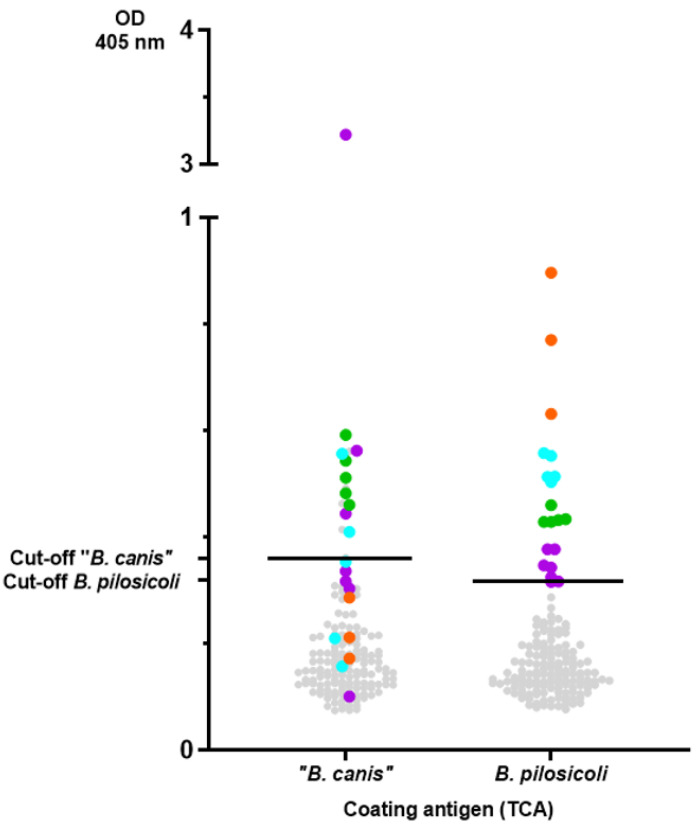
Results of the ELISA measuring IgG antibodies against detergent-extracted antigens of *B. pilosicoli* and “*B. canis*” precipitated with trichloroacetic acid in dog sera (n = 158). Cut-offs were calculated via a quantile-based approach, yielding the following cut-off values: 0.36 for “*B. canis*” and 0.32 for *B. pilosicoli*. Sera with distinct levels of antibodies against *B. pilosicoli* are colored as follows (OD_405 nm_): values ≥ 0.32 but <0.4 are colored in violet. Values ≥ 0.4 but <0.5 are colored in green, values ≥ 0.5 but <0.6 are colored in turquoise, and values ≥ 0.6 are colored in orange. The same sera are also color-coded in the “*B. canis*” results, unrelated to the calculated OD value for “*B. canis*”.

**Figure 6 vetsci-11-00302-f006:**
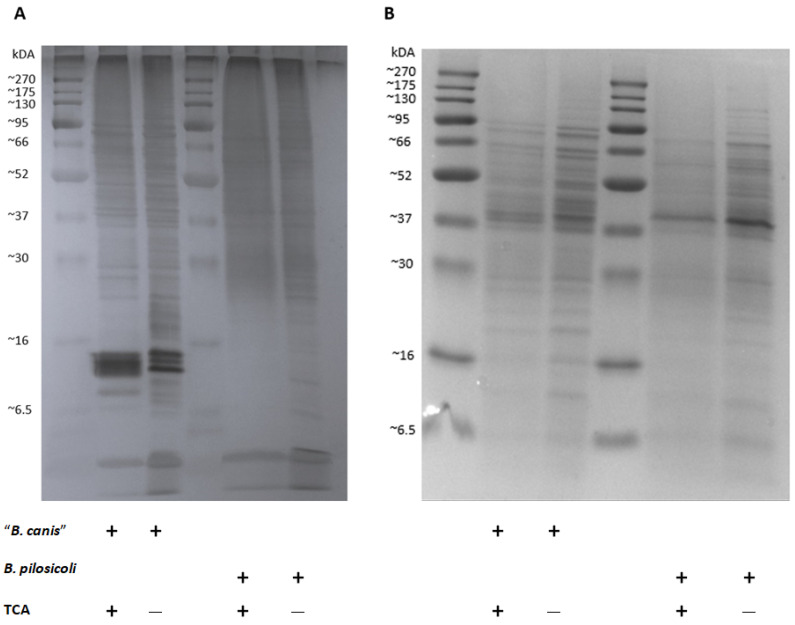
Silver staining (**A**) and Coomassie Blue staining (**B**) following SDS-PAGE of detergent-extracted *Brachyspira* antigens before and after precipitation with trichloroacetic acid (TCA).

## Data Availability

The raw data supporting the conclusions of this article will be made available by the authors upon reasonable request.

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
