# Peer review of "Detection of Serum IgG Specific for Brachyspira pilosicoli and “Brachyspira canis” in Dogs"

_vetsci, 2024, doi:10.3390/vetsci11070302_

Round 1

Reviewer 1 Report

Comments and Suggestions for Authors

This study established laboratory tests to detect specific antibodies against bacteria of the genus Brachyspira in canine sera. These bacteria can cause inflammation in the large intestine and affect various animal species, including dogs. The study focused on two species of Brachyspira isolated from canine fecal samples and rectal swabs. Different antigen preparations were evaluated, and a suitable antigen preparation was successfully identified. These laboratory tests are expected to help indirectly detect infections in dogs and to better understand the potential pathogenic role of Brachyspira in dogs.

This work is interesting, the methodology is fine, up to date, the manuscript quite well written, and the data presented also seem fine and support the conclusions what the authors made. The work presented in this manuscript is adequate. For this article, I provide some suggestions as follows:

1. How is it possible to test more serum to improve the reliability of experimental results?

3. The lack of positive or negative reference serum may not be convincing for the experiment.

2. The description of figures need to be more concise, and more content should be presented in the description of the results.

Reviewer 2 Report

Comments and Suggestions for Authors

In this manuscript, the authors describe the process used to develop an elisa test for the detection of IgG against Brachyspira pilosicoli and Brachyspira cani.

Although well written, I have some suggestions for authors.

I would suggest dividing figure 1 into two parts: b and c are results, not methods.

In the methods it is reported that 158 sera were analysed and in the supplementary material there are 168: is this an error? Why are some subjects coloured in the supplementary material? Do they correspond to the colours used in figure 4? 

Also, in the results, polyclonal anti-'B. canis' is mentioned, but in the methods there is no reference to this.

In addition, I would suggest converting figure 3 into a histogram in order to be consistent with the rest of the graphs.

The importance of using this method is unclear: if B. canis is a commensal and no pathogenesis has been demonstrated, why should this test be useful?  Perhaps the authors should discuss this aspect more.

Finally, I cannot identify a conclusion. Please expand this section.
